# Clinical Validation of an Automated Fluorogenic Factor XIII Activity Assay Based on Isopeptidase Activity

**DOI:** 10.3390/ijms22031002

**Published:** 2021-01-20

**Authors:** Martina Leitner, Christian Büchold, Ralf Pasternack, Nikolaus B. Binder, Gary W. Moore

**Affiliations:** 1Technoclone Herstellung von Diagnostika und Arzneimitteln GmbH, Brunner Str. 67, 1230 Vienna, Austria; martina.leitner@technoclone.com (M.L.); nikolaus.binder@technoclone.com (N.B.B.); 2Zedira GmbH, Roesslerstrasse 83, D 64293 Darmstadt, Germany; buechold@zedira.com; 3Department of Haematology, Specialist Haemostasis Unit, Cambridge University Hospitals NHS Foundation Trust, Hills Road, Cambridge CB2 0QQ, UK; 4Faculty of Science and Technology, Middlesex University, The Burroughs, Hendon, London NW4 4BT, UK

**Keywords:** automation, bleeding disorder, blood coagulation, clinical validation, diagnostic assay, factor XIII, isopeptidase, transglutaminase

## Abstract

Hereditary factor XIII (FXIII) deficiency is a rare autosomal bleeding disorder which can cause life-threatening bleeding. Acquired deficiency can be immune-mediated or due to increased consumption or reduced synthesis. The most commonly used screening test is insensitive, and widely used quantitative assays have analytical limitations. The present study sought to validate Technofluor FXIII Activity, the first isopeptidase-based assay available on a routine coagulation analyser, the Ceveron s100. Linearity was evidenced throughout the measuring range, with correlation coefficients of >0.99, and coefficients of variation for repeatability and reproducibility were <5% and <10%, respectively. A normally distributed reference range of 47.0–135.5 IU/dL was derived from 154 normal donors. Clinical samples with Technofluor FXIII Activity results between 0 and 167.0 IU/dL were assayed with Berichrom^®^ FXIII Activity, a functional ammonia release assay, and the HemosIL^™^ FXIII antigen assay, generating correlations of 0.950 and 0.980, respectively. Experiments with a transglutaminase inhibitor showed that Technofluor FXIII Activity can detect inhibition of enzymatic activity. No interference was exhibited by high levels of haemolysis and lipaemia, and interference by bilirubin was evident at 18 mg/dL, a level commensurate with severe liver disease. Technofluor FXIII Activity is a rapid, accurate and precise assay suitable for routine diagnostic use with fewer interferents than ammonia release FXIII activity assays.

## 1. Introduction

Factor XIII (FXIII) is a protransglutaminase whose activated form (FXIIIa) is the final enzyme in the sequence of blood coagulation biochemistry and plays a key role in clot stabilization and maturation [1,2]. In plasma, FXIII circulates as a non-covalent, zymogen heterotetramer (FXIII-A_2_B_2_) comprising two catalytic A-subunits and two inhibitory/carrier B-subunits [3,4]. The A-subunits are produced by monocyte/macrophage cell lines and the B-subunits are synthesised and secreted by hepatocytes [4,5]. The FXIII-A_2_B_2_ complex circulates in the plasma bound to fibrinogen 2 [6,7]. The cellular form of FXIII, such as that found in the platelet cytoplasm, is a homodimer of the A-subunits identical to those in plasma.

In contrast to the substrate cleaving functions of other enzymes of coagulation, FXIIIa catalyses the formation of new covalent bonds [8]. It exhibits transglutaminase activity that introduces ε-(γ-glutamyl)lysyl isopeptide bonds into single protein substrates, such as fibrin, and can also cross-link different proteins to each other [4]. The driving force of the reaction is the concomitant release of ammonia. Activation of plasma FXIII is achieved through thrombin cleavage of the Arg37-Gly38 peptide bond on each of the A-subunits, resulting in release of an N-terminal activation peptide [9]. This consequently destabilises the interaction between A-subunits and B-subunits whereupon binding of calcium ions to high affinity Ca^2+^ binding sites on the A-subunits causes the B-subunit to dissociate. In the presence of Ca^2+^, the cleaved FXIII-A_2_ undergoes conformational changes to expose the respective active site cysteine. In an orchestrated sequence, FXIIIa covalently cross-links abutting fibrin γ-chains and α-chains, thereby providing mechanical stability to the fibrin network. In parallel, FXIIIa cross-links α_2_-antiplasmin to fibrin, rendering the clot biochemically stable and thus preventing premature fibrinolysis by plasmin [5,10]. FXIII also plays a crucial role in wound healing by cross-linking extracellular matrix proteins and by promoting cellular signalling in leukocytes and endothelial cells [11].

Inherited FXIII deficiency is a rare, autosomal recessive bleeding disorder with a worldwide incidence of approximately one per 1–3 million [12]. Severe deficiency can cause life-threatening bleeding, with intracranial bleeding being the major cause of death [13]. The most common and often first symptoms of FXIII deficiency are delayed umbilical cord bleeding and soft tissue haematoma. Almost all cases are deficiencies of FXIII-A, and even severe deficiencies of FXIII-B present with milder bleeding symptoms than with FXIII-A deficiencies. FXIII-A deficiency is also frequently associated with impaired wound healing and recurrent miscarriage. Heterozygosity can result in bleeding upon challenge, and acquired deficiency has been reported due to neutralising or clearing antibodies and consumption or decreased synthesis related to major surgery, pulmonary embolism, ulcerative colitis, liver cirrhosis, sepsis and disseminated intravascular coagulation [12,13].

Since cross-linking of fibrin is not required to detect fibrin formation in clot-based assays, standard screening tests for coagulation disturbances (prothrombin time, activated partial thromboplastin time, thrombin time and even the Clauss fibrinogen activity assay) are incapable of detecting FXIII deficiencies and different assays have been developed. The current diagnostic armamentarium for detecting FXIII deficiencies has been the subject of recent reviews detailing the advantages and limitations of available assays, with Durda et al. concluding that a reliable high-throughput method to directly measure FXIII activity is needed to facilitate the International Society on Thrombosis and Haemostasis (ISTH) scientific sub-committee (SSC) recommendations for diagnosis and classification of FXIII deficiency [12,13,14,15]. Clot solubility screening tests remain widely used, but whilst they are simple and inexpensive, they are not standardised or quantitative and only detect severe reductions in FXIII. Commercially available quantitative assays measuring FXIII activity or antigen are more reliable and standardised but users must be aware of limitations of each assay type, and not all are available in automated format. We recently reported the development and technical performance of a new, fluorogenic isopeptidase-based FXIII activity assay, the only assay of that type that is automatable on a routine coagulation platform, the Ceveron s100 (Technoclone, Vienna, Austria) [16]. The present study extends that work by assessing clinical performance of the commercially available form of the assay, Technofluor FXIII Activity (Technoclone, Vienna, Austria), on the Ceveron s100 coagulation analyser.

## 2. Results

### 2.1. Linearity

The SSC/ISTH Secondary Coagulation Standard Lot #5 has a calibrated FXIII activity value of 77.0 IU/dL. The linearity plot of dilutions in FXIII-deficient plasma shown in Figure 1a reveals linearity in the range 2.3 to 77.0 IU/dL, with a correlation coefficient of >0.99. The recovery calculated between the measured and expected values of the sample dilutions within that range was between 91% and 103%. The measured FXIII activity levels in the two normal plasma samples were 132.6 and 88.6 IU/dL. Their linearity plots are shown in Figure 1b,c respectively, revealing linearity down to 4.4 and 3.1 IU/dL, respectively, both with correlation coefficients of >0.99. We previously reported a lower limit of detection (LLoD) of 0.4 IU/dL and a lower limit of quantification (LLoQ) of 0.9 IU/dL [16].

### 2.2. Repeatability and Reproducibility

Coefficients of variation for the repeatability and reproducibility of the six samples spanning reduced, borderline and normal FXIII levels assayed with Technofluor FXIII Activity on Ceveron s100 analysers are presented in Table 1.

### 2.3. Reference Range

The population distribution for the 154 normal donor samples was Gaussian, generating a Shapiro–Wilk *p*-value for normality of 0.7133 (Figure 2). No outliers were detected by Tukey assessment [17]. Calculated as ±2 standard deviations of the mean, the reference range for Technofluor FXIII Activity on the Ceveron s100 was 47.0–135.5 IU/dL (0.47–1.36 IU/mL).

### 2.4. Clinical Samples

The 110 clinical samples had FXIII activity values spanning the range 0–167.0 IU/dL when assayed with Technofluor FXIII Activity on the Ceveron s100. Comparison of the results with those obtained from assaying the samples with the Berichrom^®^ FXIII assay, the functional ammonia release assay, generated a Passing–Bablok regression fit of y = −5.448 + 1.011x, with a correlation of 0.950. A two-tailed paired t-test confirmed no significant population differences, *p* = 0.252201. Compared to the widely used HemosIL FXIII antigen assay, a Passing–Bablok regression fit of y = −1.184 + 1.124x was obtained, with a correlation of 0.980. A two-tailed paired t-test confirmed no significant differences, *p* = 0.399378. Correlation plots are shown in Figure 3.

Whilst the correlation between Technofluor FXIII Activity and Berichrom^®^ FXIII Activity was good, five of the normal samples generated Technofluor FXIII Activity results that were >10% higher than with Berichrom^®^ FXIII Activity, the latter being concordant with the HemosIL FXIII antigen values in four of the five samples. The Technofluor/Berichrom^®^/HemosIL (IU/dL) results, respectively, were 165.9/146.3/136.8, 147.6/126.0/120.8, 100.1/89.0/99.7, 134.6/115.9/119.1 and 100.5/82.8/82.0. Sixteen of the 63 patient samples generated Technofluor FXIII Activity results that were >10% lower than with Berichrom^®^ FXIII Activity, with 14 having concordant Technofluor and HemosIL results. Results with Technofluor FXIII Activity ranged from 2.8 to 69.4 IU/dL (mean/median 42.5/39.7), the range with Berichrom^®^ FXIII Activity was 13.1–85.4 (mean/median 55.1/52.3) and the range with HemosIL FXIII Antigen was 3.0–82.0 (mean/median 40.7/36.5). Percentage differences between Technofluor FXIII Activity and Berichrom^®^ FXIII Activity ranged from 10.3 to 17.7 (mean/median 13.2/12.8). Despite the numerical discrepancies, the different lower limits of normal (Technofluor/Berichrom^®^ 47.0/70.0 IU/dL) meant that only three samples had a discordant diagnosis in terms of whether FXIII activity was reduced or not. The Technofluor/Berichrom^®^/HemosIL results (IU/dL), respectively, were 53.5/66.3/48.3, 47.5/60.2/36.6 and 52.8/67.6/53.1, where the Berichrom^®^ values were out of step with the others. One patient sample with severely reduced Technofluor/HemosIL results (IU/dL), respectively, of 2.0/3.0 gave an overestimation of activity, 13.1 IU/dL, with the Berichrom^®^ assay.

Other patient samples of note were two with Technofluor/Berichrom^®^/HemosIL results (IU/dL), respectively, of 38.4/92.3/36.4 and 24.0/60.2/22.9, where Technofluor and HemosIL results were concordantly reduced whilst the Berichrom^®^ values were more than double those with the other two assays, one of them within the reference range. A third sample with discrepant results gave respective values of 27.2/53.8/42.9. Results from the known FXIII-deficient ECAT samples for each assay are shown in Table 2.

### 2.5. Interferences

Figure 4 shows FXIII levels of the normal sample with Technofluor FXIII Activity and HemosIL FXIII Antigen in the presence and absence of the T101 FXIII inhibitor molecule. The antigenic level remained constant whilst the activity level fell with increasing T101 concentration.

The effects of 1 mM ammonia on the measurement of different levels of FXIII activity with the Technofluor isopeptidase FXIII activity assay and the Berichrom^®^ ammonia release FXIII activity assay are shown in Figure 5. There was negligible effect of that level of ammonia on the Technofluor assay but FXIII activity was increasingly overestimated at decreasing FXIII activities, shown as an increase in percentage recovery.

Figure 6 shows the effect of elevated fibrinogen levels on the Technofluor isopeptidase FXIII activity assay and the Berichrom^®^ ammonia release FXIII activity assay. The elevated fibrinogen levels had negligible effect on the baseline FXIII activity level of 3.1 g/L with the Technofluor assay but the percentage FXIII recovery increased with elevated fibrinogen levels with the Berichrom^®^ assay.

No interference in the Technofluor FXIII Activity assay was found with haemolysis up to the maximum level of haemoglobin measured of 500 mg/dL, up to 18 mg/dL of conjugated bilirubin and up to the maximum level of intralipid measured of 1400 mg/dL.

## 3. Discussion

Detection of FXIII deficiency is an essential component of the analytical repertoire in a diagnostic haemostasis laboratory. Resource restrictions can limit diagnostic efficacy when insensitive clot solubility assays are the only available option, and screening with automated antigenic assays precludes detection of qualitative defects [12,13,14,15]. Consequently, current ISTH-SSC guidelines recommend initial screening with a quantitative, functional FXIII assay, the most commonly employed and widely available being ammonia release assays, which are easily automated. Although the main drawback of ammonia release assays can be overcome by use of an iodoacetamide blank, it is not supplied with every commercial kit version of the assay and there is variable implementation of the blanking step when not supplied by the manufacturer [18]. Omission of the blank can lead to overestimation of FXIII activity by as much as 2–15 IU/dL and precipitate serious clinical consequences if a deficiency goes undetected and therefore not correctly diagnosed or managed [5,14]. Isopeptidase assays for FXIII activity are rapid, direct, automatable and independent of subsequent enzymatic steps and the present study builds on our previous report [16], in order to validate the Technofluor FXIII Activity assay with clinical samples against an assay/platform-specific reference range, in comparison with two widely used assays.

We previously reported that the novel substrate in the Technofluor FXIII Activity assay is only dependent on thrombin-activated FXIII, rendering the assay highly specific [16]. Accuracy was evidenced with samples of known FXIII activity levels, with all samples achieving >95% recovery of target values. The present study confirms assay linearity with two normal samples in addition to employing the SSC/ISTH Secondary Coagulation Standard Lot #5 [16]. Since clinically significant bleeding episodes occur with FXIII activity levels up to 3.0 IU/dL [14], the LLoD and LLoQ of 0.4 and 0.9 IU/dL, respectively, indicate that the assay can detect such severely reduced levels. Coefficients of variation for repeatability within runs were <5% and <10% for reproducibility between runs, for samples across the measuring range, both values being below the 15% acceptability criterion in the European Medicines Agency guideline on bioanalytical method validation [19]. 

A reference range specific to the performance of the Technofluor FXIII Activity assay on the Ceveron s100 analytical platform was generated from a large number of normal donor plasmas (n = 154). Visual inspection of the population distribution, as shown in Figure 2, revealed a Gaussian distribution, with the Shapiro–Wilk *p*-value of 0.7133 confirming no significant deviation from normality. Consequently, the reference range was derived parametrically as ±2 standard deviations of the mean [17]. Interestingly, the reference range of 47.0–135.5 IU/dL maps closely to the clinical range of 50–150 IU/dL employed diagnostically for most coagulation factors rather than the manufacturer ranges for Berichrom^®^ FXIII activity of 70–140% of normal and HemosIL™ FXIII antigen of 75.2–154.8%. The typical bleeding symptoms of severe FXIII deficiency tend to manifest at levels of <3–5% [5] and heterozygotes do not experience significant spontaneous haemorrhage with levels above 30% [20], whilst bleeding has been reported in acquired FXIII deficiency at levels between 5 and 40% [21]. Having a lower reference range limit that better reflects FXIII levels where patients are symptomatic has potential to improve diagnostic accuracy and reduce overdiagnosis.

Comparing results from clinical samples with FXIII levels spanning severe, moderate and mild reductions and normality, when assayed with the three types of assays, generated very good overall correlations. Despite the correlation between the two activity assays of 0.950, there were some samples where the results were >10% apart. Five of the normal samples with Technofluor FXIII Activity results >100 IU/dL were accompanied by lower Berichrom^®^ FXIII activity values that were concordant with the antigenic levels. This could suggest reduced linearity with Technofluor FXIII Activity at higher FXIII concentrations, yet the correlation coefficient for the linearity plot of the normal sample with FXIII activity of 132.6 IU/dL was >0.99. Conversely, 16 patient samples with FXIII levels <100 IU/dL generated Technofluor FXIII Activity results that were 10.3–17.7% lower than with the Berichrom^®^ FXIII activity assay, most of the former being concordant with the antigenic level. The different lower limits of normal meant that all but three were diagnostically equivalent results, the Berichrom^®^ FXIII activity results with those three being discordant with the other two assays. The experiment with 1mM ammonia did show increasing interference with decreasing FXIII level, so this is a possible explanation for these results. Unfortunately, there were no further aliquots available of the patient samples to test for interferents. Whilst many of these discrepancies may have been minimised or eliminated by blanking the ammonia release assay, it is important to note that the data serve to compare the Technofluor FXIII Activity assay with the widely used Berichrom^®^ FXIII activity assay in the form it is available in commercially. Three other samples from the patient cohort generated markedly discrepant results. Two had concordantly reduced Technofluor FXIII activity and HemosIL FXIII antigen results, indicating quantitative deficiencies, yet the Berichrom^®^ FXIII activity results were incongruously higher than the antigenic results. One possible explanation is that the samples had high ammonia levels that were falsely elevating results from the ammonia release assay, although the discrepancies of 53.9% and 36.2% against the Technofluor FXIII activity levels were more marked than the up to 15% that is normally attributed to ammonia interference at lower levels than these, so they may have been merely discrete analytical errors or due to additional interferents. The patient sample with Technofluor/Berichrom^®^/HemosIL results (IU/dL), respectively, of 2.0/13.1/3.0 was most likely an example of overestimation of severe deficiency due to ammonia interference. Taking an FXIII activity/FXIII antigen ratio of <0.6 (when the FXIII activity is below the reference range) as indicative of a qualitative deficiency [22], the Technofluor FXIII activity/HemosIL antigen ratio of 0.63 in the third sample was borderline for qualitative deficiency, whilst the Berichrom^®^ FXIII activity/HemosIL antigen ratio was 1.25, indicative of a quantitative deficiency. A possible explanation here is that the Berichrom^®^ FXIII activity result was reduced because the actual FXIII concentration (i.e., antigen) was reduced, whilst the Technofluor FXIII Activity was additionally detecting a functional defect to which the Berichrom^®^ FXIII activity assay is insensitive. This is akin to different functional defects of antithrombin and protein C that manifest in activity assays based on one principle but not another [23,24]. There was broad between-assay agreement with the seven low-FXIII samples from ECAT surveys, although two of them, Sample 3 and Sample 4, had concordant Technofluor FXIII Activity and HemosIL FXIII antigen results (IU/dL) of 7.9/7.3 and 6.1/6.5, respectively, whilst the Berichrom^®^ FXIII Activity results were lower at 1.3 and 0.3, respectively. This could be due to the Berichrom^®^ assay being more accurate where levels are close to zero, yet two other samples with very low levels were ostensibly concordant with all assays. An alternative explanation is that Sample 3 and Sample 4 had low fibrinogen levels, which can falsely reduce FXIII activity results in ammonia release assays. Clauss fibrinogen activity assays using the Fibrinogen Reagent Kit (Technoclone, Vienna, Austria) on the Ceveron s100 showed that Sample 3 and Sample 4 had markedly reduced levels of 0.31 and 0.46 g/L, respectively (reference range 1.8–4.5 g/L), suggesting this was indeed the cause of the discrepancies. A limitation of the study is that no clinical information was available on the patient samples to correlate measured FXIII activity levels with bleeding. However, the highly concordant Technofluor FXIII Activity and HemosIL FXIII antigen data do show that the Technofluor assay can distinguish between severe, moderate and mild deficiencies and normality.

Integral to the diagnostic work-up upon finding a reduced FXIII level in a setting of possible acquired FXIII deficiency is the ability to detect functional inhibition of FXIII. In the absence of clinical samples containing such antibodies (less than 100 cases have been described up to 2017 [12]), we challenged the Technofluor FXIII Activity assay with a small-molecule transglutaminase inhibitor as an antibody surrogate. FXIII activity was reduced in a dose-dependent manner, indicating the assay is susceptible to inhibition and suggesting it can play a diagnostic role in detection of immune-mediated acquired FXIII deficiency. 

Given the variable implementation of the blanking step for ammonia release assays, the lack of ammonia interference in the Technofluor FXIII Activity assay represents a valuable improvement over ammonia release assays, and similarly for the lack of interference by elevated fibrinogen. No significant interference was found for haemolysis and lipaemia. A normal level for conjugated bilirubin is <0.3 mg/dL and jaundice/icterus is usually evident when total bilirubin exceeds 2.5 mg/dL. Thus, the level of 18 mg/dL at which conjugated bilirubin was shown to interfere with the assay would only be experienced in severe liver disease.

Accurate detection of FXIII deficiency and the degree of reduction are vital to effective diagnosis and management of FXIII deficiency. Additionally, there is increasing recognition that better detection of non-autoimmune-acquired FXIII deficiency can prompt appropriate treatment to improve clinical outcomes [25,26], so availability of a high-throughput FXIII activity assay automated on a routine coagulation analyser could facilitate more regular testing beyond investigating for hereditary deficiencies. The Technofluor FXIII Activity assay based on isopeptidase activity as performed on the Ceveron s100 analyser is a rapid, accurate and precise functional assay of FXIII with analytical advantages over the commonly used clot solubility and ammonia release assays. 

## 4. Materials and Methods

### 4.1. Technofluor FXIII Activity Assay

The Technofluor FXIII Activity assay (Technoclone, Vienna, Austria) capitalises on the FXIIIa-dependent hydrolysis of γ:ε isopeptide bonds by employing an FXIII substrate with a fluorophore and a quencher that is linked by a carboxamide bond [27]. The assay employs a novel substrate, A138 (Zedira, Darmstadt, Hesse, Germany), because the original substrate for the FXIII isopeptidase-based assay, A101 (Zedira, Darmstadt, Hesse, Germany), was not compatible with the Ceveron platform optics regarding excitation and emission wavelengths [16].

Briefly, FXIII is activated by thrombin in the presence of calcium and the isopeptidase activity of the FXIIIa cleaves the side-chain carboxamide bond from a modified amino acid substrate, thereby releasing the dark quencher (2,4-dinitrophenyl) linked to the cadaverine spacer. The consequent increase in fluorescence results from the unmasking of an N-terminally attached fluorophore, N-Methyl-2-aminobenzoic acid (N-Me-Abz). The enzymatic activity of FXIIIa directly translates into an increasing signal over 30 min [16]. 

All reagents in the Technofluor FXIII Activity assay kit were provided as lyophilised material. Immediately prior to use, the substrate and thrombin reagent were reconstituted in 2.0 mL of distilled water and left to equilibrate for 30 min at room temperature. The assay buffer, supplied ready for use, was also brought to room temperature for 30 min before use. Citrated plasma samples were diluted in equal volumes on the Ceveron s100 (Technoclone, Vienna, Austria) in FXIII assay buffer (56 mM Tris-HCl, 115 mM NaCl, 0.14 mM PEG 8000, 5.7 mM H-Gly-OMe, and Polybrene). An amount 40 µL of the pre-dilution was then added to a cuvette preheated to 37 °C. Thereafter, 80 µL of FXIII substrate with 2,4-dinitryophenyl dark quencher (80 µM final concentration) and 80 µL of thrombin reagent (10 U/mL bovine thrombin final concentration, 1.2 mg/mL tetrapeptide polymerisation inhibitor (Gly-Pro-Arg-Pro-amide) and calcium chloride) were added consecutively. FXIII activity was measured as increase in fluorescence emission in relative fluorescence units (RFU) for 30 min after addition of the thrombin and substrate reagents. Excitation and emission wavelengths were 360 and 460 nm, respectively. ΔRFU from 7 to 22 min were used for calculation of FXIII activity against the calibrator, SSC/ISTH Secondary Coagulation Standard Lot #5 (NIBSC, South Mimms, Potters Bar, UK), which is traceable to the 1st International WHO standard for FXIII in plasma.

### 4.2. Berichrom Chromogenic FXIII Activity Assay

The Berichrom^®^ FXIII assay kit (Siemens Healthineers, Marburg, Hesse, Germany) is an automated assay based on the ammonia release principle [5]. FXIII in the sample being tested is activated by thrombin in the presence of Ca^2+^, and fibrin formed by the action of the thrombin accelerates this reaction. An inhibitory peptide prevents polymerisation of the fibrin so that a clot is not formed during the assay. Transglutaminase activity of the FXIIIa cross-links a glutamine-containing oligopeptide substrate to glycine ethyl ester, thereby releasing ammonia. The ammonia release is measured in a side reaction involving ammonia-dependent reduction of NADH, which is detected spectrophotometrically at 340 nm.

The Berichrom^®^ FXIII assay kit comprises an activator reagent containing 10 IU/mL bovine thrombin, 2 g/L clot inhibitor (Gly-Pro-Arg-Pro-Ala·amide), 1.2 g/L calcium chloride, 10 mg/L hexadimethrine bromide, bovine albumin and 100 mM BICINE buffer, which is reconstituted in 5.0 mL of a diluent containing 0.5 g/L NADH, and bovine albumin. A detection reagent containing 20 IU/mL glutamate dehydrogenase, 2.4 g/L synthetic peptide as the FXIII substrate, ADP, 1.4 g/L glycine ethylester, 2.7 g/L α-ketoglutarate, bovine albumin and 10 mmol/L HEPES buffer is reconstituted in 5.0 mL distilled water. On a Dade Behring BCT coagulation analyser (Siemens Healthineers, Marburg, Hesse, Germany), 18 µL of citrated plasma sample was added to 180 µL of an equal volume mixture of activator and detection reagent. Change in absorbance during NAD reduction at 304 nm was measured over 300 s. The assay was calibrated with Standard Human Plasma (Siemens), which is traceable to WHO coagulation standards. The manufacturer’s reference range is given as 70–140% of normal.

### 4.3. HemosIL™ FXIII Antigen Assay

The HemosIL™ FXIII antigen (Ag) assay (Instrumentation Laboratory, Bedford, MA, USA) is an automated immunoturbidimetric assay for quantifying FXIII concentration irrespective of function [14]. The principle involves polystyrene latex particles of uniform size that are coated with polyclonal rabbit antibodies that are highly specific for FXIII-A [28]. Addition of the latex reagent to plasma results in agglutination of the latex particles via FXIII-A in the sample, the degree of which is directly proportional to the FXIII Ag concentration. The agglutination is measured turbidimetrically by increase in light transmittance caused by the agglutinated particles.

The HemosIL™ FXIII antigen kit comprises a latex reagent containing a suspension of anti-FXIIIA-coated polystyrene latex particles and vials of TRIS buffer containing bovine serum albumin buffer. On an ACL TOP 500 CTS coagulation analyser (Instrumentation Laboratory, Bedford, MA, USA), 20 µL of citrated plasma sample was added to 120 µL of buffer and 20 µL of latex reagent. Change in absorbance due to increase in light transmittance from particle agglutination was measured at 671 nm with an acquisition time of 300 s. The assay was calibrated with HemosIL Calibration Plasma (Instrumentation Laboratory, Bedford, MA, USA), which is traceable to NIBSC coagulation standards. The manufacturer’s reference range is given as 75.2–154.8%.

### 4.4. Linearity

Citrated plasma samples from two healthy donors and the SSC/ISTH Secondary Coagulation Standard Lot #5 were each diluted in FXIII-deficient plasma (Technoclone, Vienna, Austria), an immunodepleted preparation, in various ratios, to generate levels spanning the assay measuring range. Each dilution was assayed in duplicate with the Technofluor FXIII Activity assay.

### 4.5. Repeatability and Reproducibility

Imprecision of the Technofluor FXIII Activity assay as performed on the Ceveron s100 was determined over the course of five days, with two measurement runs per day on two Ceveron s100 instruments using two lots of reagents. Imprecision was calculated for six samples spanning reduced, borderline and normal FXIII levels. Both instruments were calibrated on the first day and those calibrations were used for the duration of the experiment. An additional calibration curve was generated at the end which showed there was no shift from Day 1.

### 4.6. Reference Range

The reference range for FXIII activity was determined by measuring citrated plasma samples of 154 healthy donors, split into 9 individual runs, as the maximum number of simultaneous tests is limited to 36 on the Ceveron s100. The tests were performed on two different analysers with two lots of Technofluor FXIII Activity reagents and were calibrated using the SSC/ISTH Secondary Coagulation Standard Lot #5. 

### 4.7. Clinical Samples

Blood samples were collected into a one-tenth volume of 0.015 M (3.2%) tri-sodium citrate, centrifuged to obtain platelet poor plasma, and the plasma was stored at −70 °C. Immediately prior to analysis, samples were rapidly thawed for no longer than 5 min at 37 °C to prevent cryoprecipitate formation and mixed by multiple, gentle inversions. Sixty-three samples were from patients who had previously been tested for FXIII levels, although clinical data were unavailable as they were referral samples. Ten samples were obtained from normal donors to ensure samples with normal levels were included in the performance comparison with other assays. Each sample was also diluted 3:1, 1:1 and 1:3 in FXIII-deficient plasma to generate levels across the measuring range. Additionally, seven samples from patients with moderate to severe reductions in FXIII activity were supplied in lyophilised form from the ECAT external quality assurance agency.

Each sample was assayed with Technofluor FXIII Activity, Berichrom^®^ FXIII chromogenic activity assay and HemosIL™ FXIII antigen as described above to compare analytical performance of the isopeptidase assay with the two commonly employed assay types in routine diagnostic use.

### 4.8. Interferences

Acquired FXIII deficiency due to anti-FXIII antibodies is rarer still than hereditary FXIII deficiency [29] and no clinical samples were available to challenge the Technofluor FXIII Activity assay in detection of inhibition. Instead, two aliquots of a plasma with an FXIII level of 100.0 IU/dL were spiked with the transglutaminase inhibitor 1,3,4,5-Tetramethyl-2-[(2-oxopropyl)thio]imidazolium chloride (T101) (Zedira, Darmstadt, Hesse, Germany) at final concentrations of 6.25 and 12.5 µM and assayed alongside non-spiked plasma with Technofluor FXIII Activity and HemosIL™ FXIII antigen.

Ammonia release assays are prone to overestimating FXIII activity when it is present at low levels because there are other ammonia-producing and NADH-consuming reactions that occur in normal human plasma [5,14]. Consequently, these assays should be blanked with iodoacetamide, which inhibits FXIIIa [30]. Although not a theoretical interference in the isopeptidase assay, a plasma sample with FXIII activity of 100 IU/dL and dilutions of it in FXIII-deficient plasma to give 30 and 15 IU/dL were assayed with Technofluor FXIII Activity and the Berichrom^®^ FXIII chromogenic activity assay in the presence of 1 mM ammonia (Sigma-Aldrich, St. Louis, MO, USA) to assess for interference.

Elevated fibrinogen can interfere with ammonia release assays and adversely affect accuracy. To investigate whether the isopeptidase assay is affected by elevated fibrinogen, the SSC/ISTH Secondary Coagulation Standard Lot #5 was assayed neat and spiked with purified human fibrinogen (Merck, Darmstadt, Hesse, Germany) to values of 5.1 and 6.5 g/L and assayed with Technofluor FXIII Activity and Berichrom^®^ FXIII chromogenic activity assays.

Haemolysis, icterus and lipaemia are common interferences in assays for coagulation proteins. To assess the level at which haemolysis interfered with the Technofluor FXIII Activity assay, a haemolysate was prepared from anticoagulated whole blood. Plasma was removed by washing in isotonic sodium chloride, and the cells were finally resuspended in distilled water, frozen overnight at −70 °C and centrifuged to pellet debris, and dilutions of the supernatant were used to spike a plasma sample with FXIII activity of 100 IU/dL and dilutions of it in FXIII-deficient plasma to give 30 and 15 IU/dL. Concentrations of haemoglobin (mg/dL) in the supernatant dilutions were <250, 250, 375 and 500. Similarly for icterus, aliquots of the same plasmas were spiked with unconjugated bilirubin (Sigma-Aldrich, St. Louis, MO, USA) at a range of concentrations between 0.5 and 43.0 mg/dL. For lipid interference, aliquots of the same plasmas were spiked with 20% intralipid emulsion (Sigma-Aldrich, St. Louis, MO, USA) at a range of concentrations between 132 and 1400 mg/dL.

## Figures and Tables

**Figure 1 ijms-22-01002-f001:**
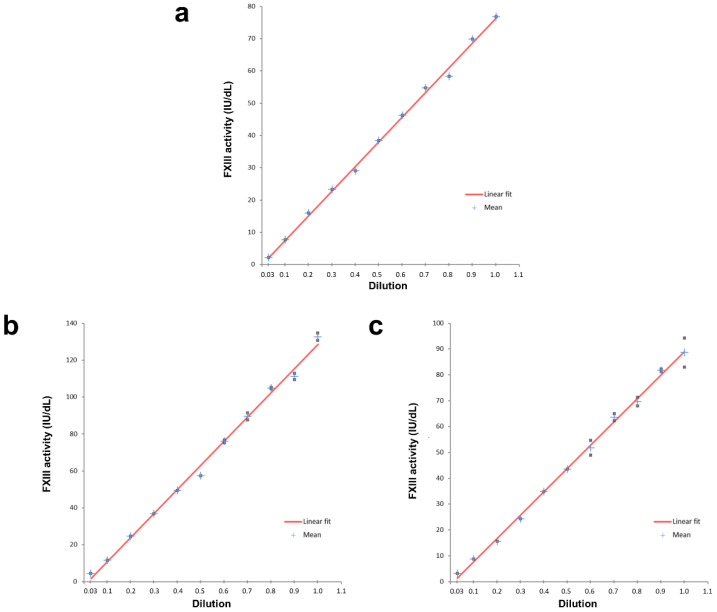
Linearity plots. (**a**) Plot for SSC/ISTH Secondary Coagulation Standard Lot #5 with a calibrated FXIII activity value of 77.0 IU/dL. (**b**) Plot for normal donor plasma with a measured FXIII activity value of 132.6 IU/dL. (**c**) Plot for normal donor plasma with a measured FXIII activity value of 88.6 IU/dL.

**Figure 2 ijms-22-01002-f002:**
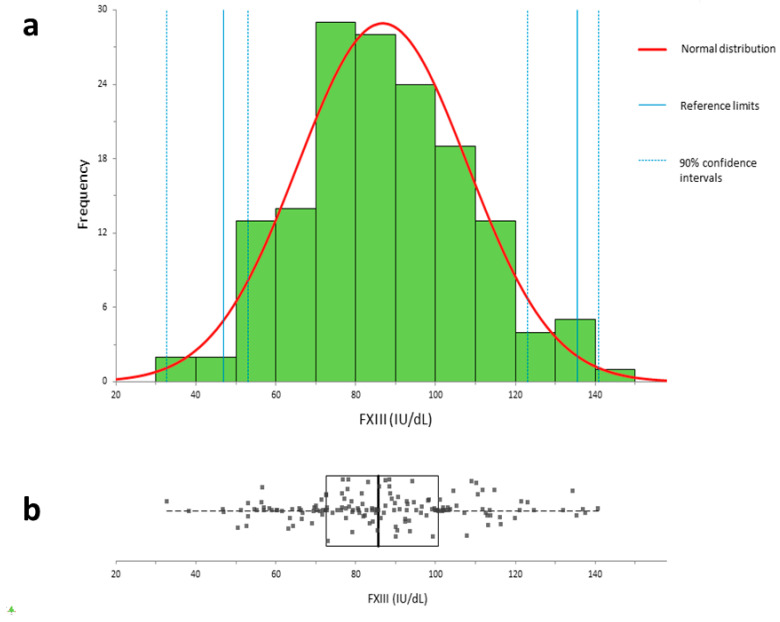
Population distribution from 154 normal donors for Technofluor FXIII Activity on the Ceveron s100. (**a**) Distribution plot (**b**) Box plot

**Figure 3 ijms-22-01002-f003:**
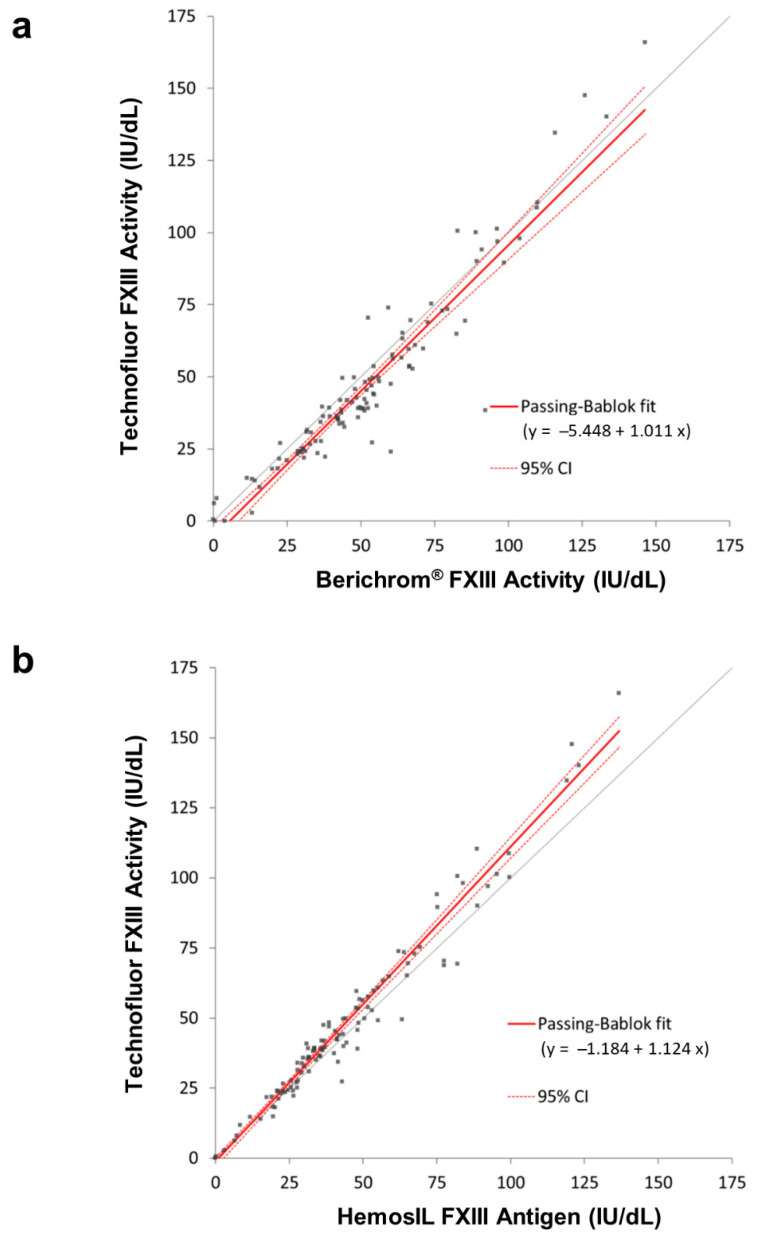
Correlation plots. (**a**) Technofluor FXIII Activity vs. Berichrom^®^ FXIII Activity. (**b**) Technofluor FXIII Activity vs. HemosIL FXIII Antigen.

**Figure 4 ijms-22-01002-f004:**
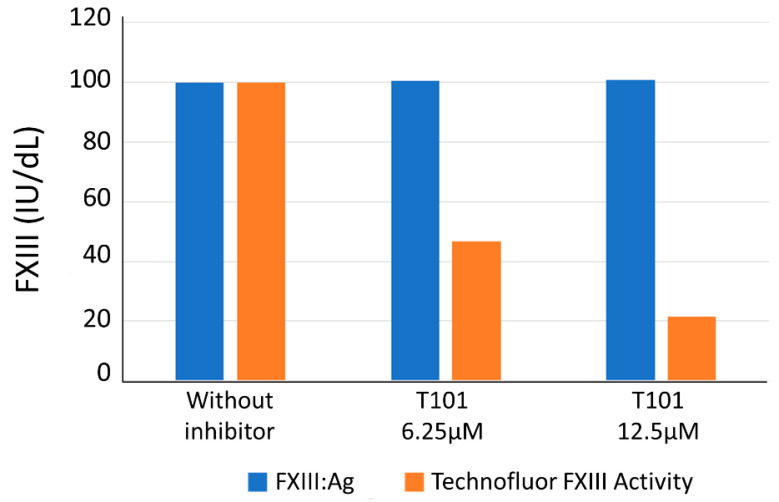
FXIII antigen and Technofluor FXIII activity in the presence and absence of a small-molecule FXIIIa inhibitor.

**Figure 5 ijms-22-01002-f005:**
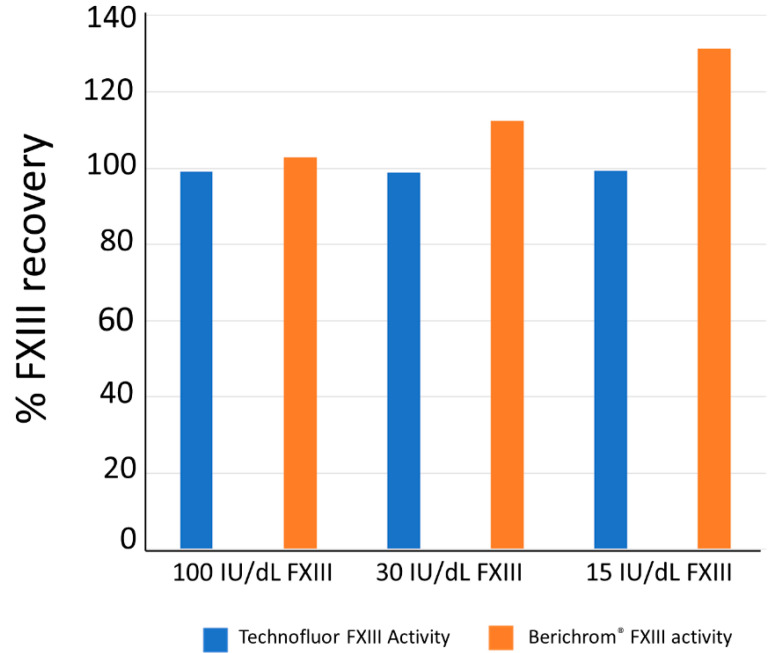
Effect of 1 mM ammonia on measurement of FXIII activity by Technofluor. FXIII Activity and Berichrom^®^ FXIII Activity assays.

**Figure 6 ijms-22-01002-f006:**
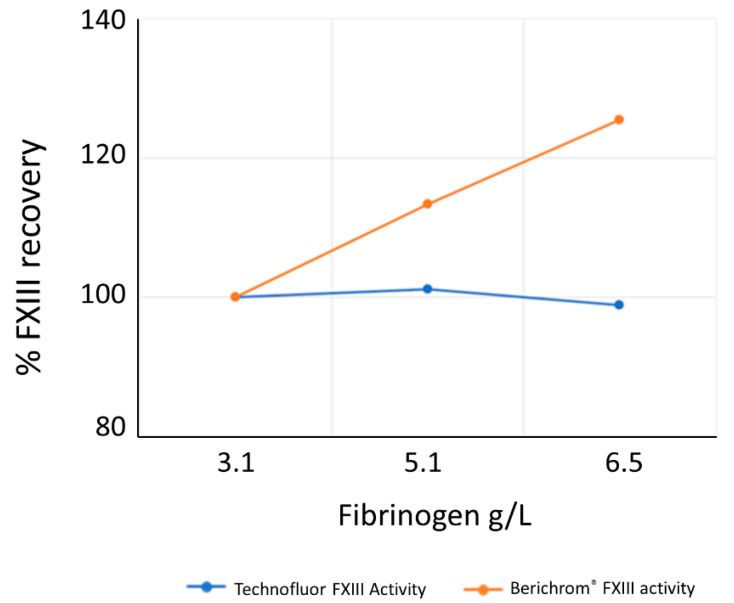
Effect of increasing fibrinogen concentration on measurement of FXIII activity by Technofluor FXIII Activity and Berichrom^®^ FXIII Activity assays.

**Table 1 ijms-22-01002-t001:** Repeatability and reproducibility.

Mean Measured FXIII Activity (IU/dL)	Repeatability CV (%)	Reproducibility CV (%)
98.2	3.8	7.6
88.1	4.2	6.3
58.3	3.7	6.0
36.7	3.1	4.1
29.3	4.8	7.5
14.7	4.9	9.2

CV, coefficient of variation.

**Table 2 ijms-22-01002-t002:** Results from known FXIII-deficient samples.

ECAT Samples	Technofluor FXIII Activity (IU/dL)	Berichrom^®^ FXIII Activity (IU/dL)	HemosIL FXIII Antigen (IU/dL)
Survey sample 1	0.0	0.6	0.0
Survey sample 2	0.5	0.0	0.3
Survey sample 3	7.9	1.3	7.3
Survey sample 4	6.1	0.3	6.5
Survey sample 5	14.9	11.4	19.6
Survey sample 6	14.0	14.1	15.4
Survey sample 7	23.1	28.5	22.2

## Data Availability

Data are available on request.

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
