# Peer review of "Clinical Validation of an Automated Fluorogenic Factor XIII Activity Assay Based on Isopeptidase Activity"

_ijms, 2021, doi:10.3390/ijms22031002_

Round 1
Reviewer 1 Report
Author report their experiences with a new method to detect levels of F XIII.
yet in their population there are “healthy donors” that in 2021 have not a clinical value and represent and abstract concept: they should specify clinical characteristics of their population in particular personal and familial history of bleeding and intake of NSAIDs , antiplatelets and anticoagulants that may alter the personal trend to bleeding and of immunosoppressive drugs that may reduce the synthesis of clotting factors.
Author Response
Thank you for your comments, for which we have the following responses:
Referee comment 1
Author report their experiences with a new method to detect levels of F XIII. yet in their population there are “healthy donors” that in 2021 have not a clinical value and represent and abstract concept:
Response
Whilst the normal donor samples were not from hospitalised patients they served the purpose of representing a population with normal levels of FXIII, and additionally, subsequent dilutions ensured that assay comparability was undertaken at different levels. Section 4.7 already states "Ten samples were obtained from normal donors to ensure samples with normal levels were included in the performance comparison with other assays. Each sample was also diluted 3:1, 1:1 and 1:3 in FXIII deficient plasma to generate levels across the measuring range" by way of describing and justifying their inclusion in the context of comparing assays with samples of differing FXIII levels.
Referee comment 2
they should specify clinical characteristics of their population in particular personal and familial history of bleeding and intake of NSAIDs , antiplatelets and anticoagulants that may alter the personal trend to bleeding and of immunosoppressive drugs that may reduce the synthesis of clotting factors.
Response
Section 4.7 already states that clinical data were unavailable on the patient samples because they were referral samples. The referee makes a valid point that clinical information would be valuable so we have added consideration of this as a study limitation to the discussion.
Additions to the manuscript are in red.
Reviewer 2 Report
The authors performed a validation study of the Technofluor FXIII activity assay using a novel isopeptidase method. This study is well-designed. The data look promising. There are several minor points.
- The Technofluor assay has very good linearity characteristics down to 3 IU/dL. It will important to also determine the lower limit of detection and quantification since most FXIII deficiencies intend to be less than 3 IU/dL.
- The authors should provide more details on the precision data. How many repeats were conducted, and were repeats performed on different days with different calibration curves?
- Sample 3 and 4 of the surveys showed the discrepancy between Technofluor FXIII activities and Brichrome FXIII activities. The authors speculated that possible hypofibrinogenemia could have caused the difference. I agree with the interpretation and I think the authors should provide the fibrinogen levels of both samples.
- Some significant discrepancies between Technofluor and Brichrome methods are obvious on figure 3a. The author should provide some explanation and investigation on the differences such as fibrinogen level, sample integrates, etc.
- Since the two methods used different FXIII enzymatic activities, is it possible that they actually could be measuring different aspects of FXiII in certain cases? Such discrepancies were observed between one-stage and two-stage factor VIII assays.
Author Response
Thank you for your positive comments and detailed review of the manuscript, for which we have the following responses:
Referee comment 1
The Technofluor assay has very good linearity characteristics down to 3 IU/dL. It will important to also determine the lower limit of detection and quantification since most FXIII deficiencies intend to be less than 3 IU/dL.
Response
We have now quoted the LLoQ and LLoD (both <1.0 IU/dL) from our previous paper in section 2.1, and have commented in the discussion that the assay is sufficient to detect deficiencies of <3 IU/dL.
Referee comment 2
The authors should provide more details on the precision data. How many repeats were conducted, and were repeats performed on different days with different calibration curves?
Response
Section 4.5 already states "Imprecision of the Technofluor FXIII Activity assay as performed on the Ceveron s100 was determined over the course of five days, with two measurement runs per day on two Ceveron s100 instruments using two lots of reagents. Imprecision was calculated for six samples spanning reduced, borderline and normal FXIII levels." We have added additional information regarding calibration in section 4.5.
Referee comment 3
Sample 3 and 4 of the surveys showed the discrepancy between Technofluor FXIII activities and Brichrome FXIII activities. The authors speculated that possible hypofibrinogenemia could have caused the difference. I agree with the interpretation and I think the authors should provide the fibrinogen levels of both samples.
Response
We have assayed the fibrinogen levels and they were both reduced, which is now in the discussion.
Referee comment 4
Some significant discrepancies between Technofluor and Brichrome methods are obvious on figure 3a. The author should provide some explanation and investigation on the differences such as fibrinogen level, sample integrates, etc.
Response
We agree that it will be valuable to expand on the discrepancies in Figure 3a rather than rely on the figure alone, thank you for the suggestion. Three additional paragraphs have been added to section 2.4 describing the discrepancies in detail, and the discussion has been extended to cover the greater detail. It also states that, unfortunately, no additional aliquots were available on the patient samples to assess for specific interferences.
Referee comment 5
Since the two methods used different FXIII enzymatic activities, is it possible that they actually could be measuring different aspects of FXiII in certain cases? Such discrepancies were observed between one-stage and two-stage factor VIII assays.
Response
This is already given consideration in the discussion in relation to the results from one of the clinical samples, as follows: "Taking a FXIII activity/FXIII antigen ratio of < 0.6 (when the FXIII activity is below the reference range) as indicative of a qualitative deficiency [22], the Technofluor FXIII activity/HemosIL antigen ratio of 0.63 in the third sample was borderline for qualitative deficiency, whilst the Berichrom® FXIII activity/HemosIL antigen ratio was 1.25, indicative of a quantitative deficiency. A possible explanation here is that the Berichrom® FXIII activity result was reduced because the actual FXIII concentration (i.e. antigen) was reduced, whilst the Technofluor FXIII Activity was additionally detecting a functional defect to which the Berichrom® FXIII activity assay is insensitive. This is akin to different functional defects of antithrombin and protein C that manifest in activity assays based on one principle but not another [23,24]."
Additions to the manuscript are in red.
Reviewer 3 Report
My main concern is that appears to be a potential that the authors have compared an optimised 'new' assay with a 'less optimised' old assay. Specifically, the authors identify a well known issue with the Behring assay, being the need to include a blank to obtain the most accurate results (discussed extensively in Discussion); however, the authors do not mention this in the methods, and so it appears they have performed the Behring assay without blanking, in which case they are comparing against a less optimised assay. Additional concern: Fig 3 shows 2 samples that are severely deficient by the Behring assay but not with the Technofluor assay; similarly, 1 sample is severely deficient by Technofluor but not Behring; are these the same samples as identified in Table 2? The first 2 may be, but I cannot see the last one there (severely deficient by Technofluor but not Behring). This latter may be an example of what may occur if a blank is not used with the Behring assay.
Author Response
Thank you for your comments, for which we have the following responses:
Referee comment 1
My main concern is that appears to be a potential that the authors have compared an optimised 'new' assay with a 'less optimised' old assay. Specifically, the authors identify a well known issue with the Behring assay, being the need to include a blank to obtain the most accurate results (discussed extensively in Discussion); however, the authors do not mention this in the methods, and so it appears they have performed the Behring assay without blanking, in which case they are comparing against a less optimised assay.
Response
The referee is correct that the Behring assay was performed without the blank and that this was not specifically stated in the methods section. The manuscript already indicates and references that there is variable implementation of the blanking step when not supplied by the manufacturer so we have added to the discussion that the data serve to compare the new assay with the widely used Behring assay in the form it is supplied commercially.
Referee comment 2
Additional concern: Fig 3 shows 2 samples that are severely deficient by the Behring assay but not with the Technofluor assay; similarly, 1 sample is severely deficient by Technofluor but not Behring; are these the same samples as identified in Table 2? The first 2 may be, but I cannot see the last one there (severely deficient by Technofluor but not Behring). This latter may be an example of what may occur if a blank is not used with the Behring assay.
Response
The referee is correct that the two samples severely deficient with Behring but not with Technofluor are those identified in Table 2, and that the other one is not in Table 2. That third sample is now specifically mentioned in the additional detail in the results and discussion that have been added in response to a comment from referee 2.
Additions to the manuscript are in red.
Round 2
Reviewer 1 Report
I like to Thank authors to develop my criticisms